# Amyotrophic Lateral Sclerosis Pathoetiology and Pathophysiology: Roles of Astrocytes, Gut Microbiome, and Muscle Interactions via the Mitochondrial Melatonergic Pathway, with Disruption by Glyphosate-Based Herbicides

**DOI:** 10.3390/ijms24010587

**Published:** 2022-12-29

**Authors:** George Anderson

**Affiliations:** CRC Scotland & London, Eccleston Square, London SW1V 1PG, UK; anderson.george@rocketmail.com

**Keywords:** amyotrophic lateral sclerosis, motor neurons, astrocytes, mitochondria, melatonin, gut microbiome, aryl hydrocarbon receptor, butyrate, glyphosate, treatment

## Abstract

The pathoetiology and pathophysiology of motor neuron loss in amyotrophic lateral sclerosis (ALS) are still to be determined, with only a small percentage of ALS patients having a known genetic risk factor. The article looks to integrate wider bodies of data on the biological underpinnings of ALS, highlighting the integrative role of alterations in the mitochondrial melatonergic pathways and systemic factors regulating this pathway across a number of crucial hubs in ALS pathophysiology, namely glia, gut, and the muscle/neuromuscular junction. It is proposed that suppression of the mitochondrial melatonergic pathway underpins changes in muscle brain-derived neurotrophic factor, and its melatonergic pathway mimic, N-acetylserotonin, leading to a lack of metabolic trophic support at the neuromuscular junction. The attenuation of the melatonergic pathway in astrocytes prevents activation of toll-like receptor agonists-induced pro-inflammatory transcription factors, NF-kB, and yin yang 1, from having a built-in limitation on inflammatory induction that arises from their synchronized induction of melatonin release. Such maintained astrocyte activation, coupled with heightened microglia reactivity, is an important driver of motor neuron susceptibility in ALS. Two important systemic factors, gut dysbiosis/permeability and pineal melatonin mediate many of their beneficial effects via their capacity to upregulate the mitochondrial melatonergic pathway in central and systemic cells. The mitochondrial melatonergic pathway may be seen as a core aspect of cellular function, with its suppression increasing reactive oxygen species (ROS), leading to ROS-induced microRNAs, thereby altering the patterning of genes induced. It is proposed that the increased occupational risk of ALS in farmers, gardeners, and sportsmen and women is intimately linked to exposure, whilst being physically active, to the widely used glyphosate-based herbicides. This has numerous research and treatment implications.

## 1. Introduction

It is generally accepted that the pathoetiology and pathophysiology of motor neuron (MNs) loss in amyotrophic lateral sclerosis (ALS) is unknown. As with other neurodegenerative conditions, such as Alzheimer’s disease and Parkinson’s disease, a plethora of biomedical studies have investigated ALS pathophysiology leading to the accumulation of data that are yet to be placed in an accepted pathophysiological framework. Preclinical models have primarily focussed on the use of mutations in Cu/Zn superoxide dismutase (SOD)1, transactive response DNA binding protein 43 kDa (TDP-43) and fused in sarcoma/translated in liposarcoma (FUS), as well as C9ORF72 repeats, which, although infrequent, are the most common ALS susceptibility genes [1]. Approximately 30 genes, with very low frequency, have also been associated with ALS risk, including Tank-binding kinase 1 (TBK1) [2] and the CCNF gene (Cyclin-F protein) [3]. Many of these susceptibility genes are associated with other neurodegenerative disorders, mostly frontotemporal dementia (FTD). Most experimental investigations of susceptibility genes in ALS preclinical models have been carried out in MNs. However, it has been long recognized that the expression of these mutations in rodent astrocytes also has detrimental effects that parallel human ALS pathophysiology, which, in 2007, the authors proposed to indicate astrocyte-specific factors/releases underpin MNs loss in ALS [4].

Amidst the confusion as to the pathophysiological underpinnings of ALS, a variety of studies have explored, and highlighted, the relevance of different cell types, including microglia [5], astrocytes [6], oligodendrocytes [7], muscles [8], Schwann cells [9], the neuromuscular junction (NMJ) [10], and the gut microbiome [11]. A recent longitudinal study investigated levels of phosphorylated TDP-43, showing it to be evident in the gut prior to neurological symptom onset by 10 years in one case [12], indicating a crucial role for alterations in the gut and gut microbiome in ALS pathoetiology, as with many other diverse medical conditions [13,14]. Pattle and colleagues found aggregates of pTDP-43 in lamina propria macrophages and dendritic cells as well as myenteric plexus ganglion, neuronal and glial cells [12]. Aggregates of pTDP-43 were also evident in lymph node parenchyma, endothelial cells, and chondrocytes, indicating wider systemic changes prior to the symptomatology of classical ALS diagnosis.

An array of immune cells have also been linked to ALS pathophysiology, including macrophages, t-helper (Th)1 and Th17 cells, activated CD4^+^ t cells, and CD8^+^ t cells, all being higher in ALS patients than in controls [15]. Natural killer (NK) cells [16], dendritic cells [17], regulatory t cells [18], mast cells, and neutrophils [19] have also been proposed to play a role in ALS symptomatology. Recent work indicates a powerful role for heightened CD8^+^ t cells and NK cells activation, in association with elevations in interferon (IFN)-γ, tumor necrosis factor (TNF)-α, interleukin (IL)-17A and granzyme B suggestive of heightened cytotoxicity of NK cells and CD8^+^ t cells in ALS, which the authors suggest drives MN loss [20]. ALS pathoetiology has classically been thought to be top-down, whilst recent work indicates a strong bottom-up influence, with the latter emphasizing the role of changes in the NMJ and muscle. Recent work has highlighted the interactions of changes in MNs with independent changes in muscle leading to NMJ dysregulation [8]. However, given the role of the gut in the pathoetiology and the wider systemic changes evident, recent work proposes ALS to be a systemic condition [21].

An array of diverse processes has also been mooted as the primary drivers of ALS symptomatology, including the truncated version of the brain-derived neurotrophic factor (BDNF) receptor, TrkB-TI [22] acid sphingomyelinase [23], ceramide [24], aryl hydrocarbon receptor (AhR) [25], kynurenine [26], tryptophan metabolism [27], melatonin [28], and circadian rhythm disruption [29]. In the FUS model, sleep and circadian disruption precede motor deficits [30], suggesting that circadian dysregulation may be prodromal to classically defined ALS, paralleling similar data in Alzheimer’s disease [31]. 

Interestingly, AhR agonists increase soluble and insoluble TDP-43 three-fold [25], suggesting that some of the early TDP-43 increases in the gut and systemically [12] may be linked to AhR agonism, including from the pro-inflammatory cytokine induction of indoleamine 2,3-dioxygenase (IDO), which converts tryptophan to kynurenine, with kynurenine activating the AhR, in association with a decrease in tryptophan availability for the serotonergic and melatonergic pathway [32,33]. See Figure 1. The AhR has differential impacts on different immune responses, such as the inhibition of NK cell and CD8^+^ t cell activation, whilst prolonging macrophage activation. This indicates how alterations in the gut may be intimately linked to the systemic changes in ALS, as in most other medical conditions, and is elaborated below.

The lack of an integrated conceptualization of ALS pathophysiology means that there is no good treatment target(s). Consequently, ALS treatments are generally very limited in efficacy, typically involving the prescription of Riluzole (to inhibit presynaptic glutamate release) [34], edaravone (primarily as an antioxidant) [35], and selective serotonin reuptake inhibitors (SSRIs) (as MNs first lost are those with most dense serotonin inputs, which show suppressed activity in ALS, with this evident both clinically and preclinically) [36]. Platelet serotonin levels also predict survival duration in ALS patients, highlighting the role of systemic serotonin regulation [37]. However, there is little impact of these medications on ALS life expectancy, with a number of clinical research groups now resorting to finding the best medications to repurpose in the absence of any guiding pathophysiological frame of reference [38]. A wide array of nutriceuticals have been proposed as beneficial in offsetting and slowing ALS-like progression in preclinical models, including green tea and its active polyphenol, epigallocatechin gallate (EGCG) [39], resveratrol, as an AhR antagonist and sirtuin-1 inducer [40], quercetin [41] and sodium butyrate [42], as a histone deacetylase inhibitor (HDACi) [43], as well as the synergistic utility of different combinations of such nutriceuticals [44].

Numerous environmental factors have also been linked to ALS pathoetiology, including exposure to an array of different metals, electromagnetic fields, repetitive or traumatic head injuries, various neurotoxins, and excessive exercise [45,46]. The repeat expansion of C9ORF72 is linked to obsessive physical exercise in behavioral variant FTD in early life [47], suggesting that repeat-C9ORF72 may be a personality characteristic relevant to ALS susceptibility. Interestingly, although variable across different epidemiological studies, some of the most common occupations associated with ALS involve physical exercise in outdoor environments, including farmers, gardeners, football and rugby players [48,49,50,51], as well as triathletes [52]. ALS risk seems correlated with years in these occupations. All of these occupations are associated with exercise in relatively manicured environments with high levels of exposure to glyphosate-based herbicides (GBH) as well as other possible pathogens, such as microorganisms in the feces and urine of domestic and farm animals [50], all of which may gain entry via the skin as well as oral and nasal cavities.

This article reviews wide bodies of data on ALS and the various cells and processes linked to ALS pathophysiology, proposing an important role for alterations in the mitochondrial melatonergic pathway, including within the gut, glia, and muscles/NMJ. It is proposed that GBH exposure is an important environmental risk factor for ALS, with effects mediated via muscles, gut microbiome, and CNS glia. MNs are an ALS pathophysiological hub for GBH-dysregulated processes across the body and brain.

## 2. ALS Pathophysiology and Glyphosate-Based Herbicides

### 2.1. Glyphosate-Based Herbicides

Glyphosate is the major active ingredient in weedkillers (herbicides), killing both broadleaf plants and many types of grass. GBH is used across the world, with most crops consumed by humans being glyphosate-resistant, making GBH an integral aspect of food production. Regular GBH exposure is associated with an increased risk of a number of health conditions, including cancer (especially non-Hodgkin’s lymphoma subtypes) [53], as supported by the results of recent USA legal trials. This will be dealt with by courts and lawyers and is not the subject of this article, which utilizes non-controversial data on GBH’s physiological effects. Many GBH effects are on epigenetic processes, which can be persistent over time and passed on to the offspring [54], as exemplified by the rodent allergic immune response across generations [55]. GBH modulates three key sites linked to ALS pathophysiology, namely the gut, muscles, and CNS, with some common and differential effects across cells/tissues/organs via impacts on mitochondrial metabolism.

### 2.2. Gut Microbiome, ALS, and Glyphosate-Based Herbicides

There is a growing interest in the role of the gut microbiome and gut permeability in ALS pathophysiology [56], including as a treatment target [57]. The gut microbiome produces a diverse array of products, including neurotransmitters and ligands for a number of important receptors, including tryptophan-derived ligands for the AhR. However, the most investigated gut microbiome products have been the short-chain fatty acids, propionate, acetate, and, especially, butyrate. Butyrate is of particular interest due to its capacity as an HDACi, and, therefore, epigenetic regulator, as well as its capacity to increase sirtuin-3 and mitochondrial function across body cells, which may involve the upregulation of the mitochondrial melatonergic pathway [58]. Gut dysbiosis is typically associated with an increase in gut permeability, thereby allowing lipopolysaccharide (LPS) efflux, leading to the activation of the innate immune responses across body cells, including in reactive cells such as immune cells and CNS glia. Consequently, gut bacteria diversity, gut microbiome-derived products, and gut permeability-linked LPS are an area of cutting-edge research across a host of diverse medical conditions, including ALS [59].

The Shikimate pathway is a series of enzymatic reactions generating folates and aromatic amino acids, namely tryptophan, tyrosine, and phenylalanine. This pathway is not evident in animal cells but is present in their gut bacteria. GBH inhibits the Shikimate pathway in mammalian gut bacteria [60]. Importantly, over 50% of human gut bacteria are sensitive to GBH at legally allowed concentrations [61], suggesting wider impacts on the patterning and fluxes of the gut microbiome. Interestingly, data in the broiler hen gut microbiome suggest that many gut bacteria quickly become genetically resistant to GBH. However, when exposure to GBH is terminated, there is a problem for gut bacteria in adapting to the lack of GBH [62]. This indicates that GBH has distinct gut effects during exposure and following its cessation. GBH withdrawal may then allow a free ecological niche to emerge for other bacteria, with possible impacts on the gut microbiome α-diversity [62]. Exposure to a range of different herbicides can also dramatically change the fungal composition of the gut, which makes up approximately 7% of the gut microbiome [63]. As gut fungi are in intimate interaction with gut bacteria composition, wider gut microbiome changes can emerge following herbicide exposure. There is no data to date to indicate that GBH decreases levels of the short-chain fatty acids produced by the human gut microbiome, although the suppression of the Shikimate pathway is associated with a decrease in butyrate-producing bacteria in preclinical models. GBH impacts on human gut bacteria, and butyrate production, will be important to determine. GBH decreases tight junctions, thereby increasing gut permeability in preclinical models [64], in turn driving raised levels of circulating LPS, which is typically intimately linked to suppressed gut butyrate production. Overall, GBH effects on the Shikimate pathway, gut dysbiosis, and gut permeability have important consequences for CNS and systemic cell regulation.

As alterations in the gut microbiome can significantly impact the functioning of all body cells, including reactive immune and glial cells, it will be important to clarify in humans and preclinical models whether GBH effects in astrocytes and other brain cells, as well as in muscles and mitochondria more widely, are at least partly regulated by GBH-induced changes in the gut microbiome and gut permeability. For example, although it is clear that the vast majority of the body’s requirement for tryptophan is supplied via dietary intake, the suppression of the Shikimate pathway’s tryptophan production may be of relevance, particularly in cases where there is an increase in pro-inflammatory cytokine-induction of indoleamine 2,3-dioxygenase (IDO), leading to increased conversion of tryptophan to kynurenine [32], as occurs in ALS [65]. (See Figure 1). As well as being necessary for the induction of the tryptophan/serotonin/N-acetylserotonin (NAS)/melatonin pathway, tryptophan is also converted by tryptophan decarboxylase to tryptamine, which activates the AhR in intestinal epithelial cells to maintain the gut barrier. Shikimate pathway-derived tryptophan will also be utilized by intestinal epithelial cells and other gut-associated cells for the synthesis of serotonin and melatonin, both being crucial regulators of gut function, with an increase in the gut melatonin/serotonin ratio an important determinant of gut inflammation [66]. GBH effects on astrocytes are highlighted in this article. However, as with all currently conceived medical conditions associated with alterations in astrocyte interactions with neurons, wider systemic processes are intimately involved, including the gut, circadian, and immune system. The astrocyte-like enteric glial cells are a recently recognized aspect of gut regulation [67], and their interactions with gut microbiome products and vagal inputs will be important to determine in future research, including any direct and indirect GBH effects.

Alterations in specific gut bacteria in ALS and preclinical models are the subject of intense investigation, given data indicating the dramatic clinical utility of targeting the gut microbiome in ALS patients [57]. In humans, the shikimate pathway seems mainly achieved by Akkermansia muciniphila [68], which, in poultry, is significantly decreased by GBH and is not restored after GBH cessation [62]. As a different response to Akkermansia muciniphila can occur in rodents [60], investigation of GBH effects on the human, especially ALS patients’, the Shikimate pathway, and Akkermansia muciniphila will be important to determine. It should be noted that GBH does not seem deleterious to all gut bacteria, with some evidence that it can be associated with an increase in α-diversity [60]. However, data in chickens show GBH to decrease all short-chain fatty acids, with a maintained loss of butyrate and propionate after GBH cessation [62]. These authors suggest that it may be relatively easy for gut bacteria to develop genetic resistance to GBH, finding it harder to adapt to its loss, as can be the case in antibiotic resistance [62]. Generally, preclinical data indicate significant GBH impacts on the gut microbiome-derived products and gut permeability, including via effects on Shikimate pathway suppression, which clearly requires investigation in humans.

Overall, the gut microbiome and gut permeability seem to be important aspects of ALS pathophysiology and a potential treatment target. GBH effects in the gut are likely to have important consequences for the changes occurring in other organs and tissues, including the brain and muscle, as well as how these organs and tissues interact with one another via various fluxes, vesicles, and exosomes. Such alterations in intercellular communication may be placed in the context of gut bacteria interacting with and regulating, their evolutionary-derived distant relatives in the form of mitochondria across body cells and tissues [13].

### 2.3. Glia, ALS, and Glyphosate-Based Herbicides

ALS, as with many neurodegenerative conditions, is associated with astrocyte and microglia activation, with these reactive cells being the major inducers of pro-inflammatory activity in the CNS. Classical conceptualizations of brain function have been very neuron-centric, which has changed over the past two decades, especially as data have emerged on the importance of astrocytes in the regulation of neurotransmitter release as well as neuronal survival, oxidant status, and mitochondrial function. Neuronal activity has now been conceptualized as a form of immune-to-immune communication, with glia-to-glia communication a subset of this [69]. Astrocytes are an integral aspect of ALS pathophysiology, as with other neurodegenerative conditions [70], and are powerfully regulated by their interactions with microglia as well as various products passing over the blood-brain barrier (BBB).

There is a growing interest in GBH effects in the CNS, including in the pathoetiology of Parkinson’s disease [71,72] and Alzheimer’s disease [73], as well as ALS [74]. Although classically associated with Alzheimer’s disease, all these conditions show raised amyloid-β levels. GBH increases TNF-α and amyloid-β in the CNS [73]. As indicated by GBH effects in the gut, GBH will have both direct and indirect effects on CNS cells, leading to activation of the (HMGB1/hsp70/LPS)-TLR4-NF-kB-YY1 pathway, which increases beta-site amyloid precursor protein cleaving enzyme 1 (BACE1) and amyloid-β production, including in astrocytes. This may be at least partly determined by GBH induction of gut permeability, leading to increased circulating LPS, as shown in preclinical models [64]. Heightened amyloid-β levels are evident in and around MNs in the process of being lost in ALS [75], with amyloid-β aggregation enhanced by its binding to mitochondrial (mt)SOD1 [76]. This indicates a maintained inflammatory-NF-kB-YY1 activation of glia and MNs. Such pro-inflammatory activation in glia would normally be dampened by the capacity of NF-kB and YY1 to upregulate the melatonergic pathway, as shown in microglia, macrophages, and the retina [77,78,79], thereby resolving inflammation and returning astrocytes and microglia to a more quiescent state. Consequently, the maintained reactive states in glia may arise from the suppression of the melatonergic pathway following NF-kB and YY1 activation. (See Summary Figure 2). This may be an important aspect of ALS pathophysiology and will significantly impact the capacity of astrocytes to maintain their homeostatic functional regulation of neurons, including MNs. ALS susceptibility genes, such as mtSOD1, further limit the capacity of astrocytes to maintain neuronal function, oxidant status, and survival [80]. Astrocytes are crucial determinants of neuronal, including MNs, survival and are an important hub in ALS pathophysiology, indicating the importance of astrocytes for the direct and indirect effects of GBH.

As indicated above, gut microbiome-linked modulation of astrocytes will alter MNs regulation by astrocytes. Butyrate optimizes mitochondrial function in astrocytes [81], although this may be at least partly dependent upon the capacity of astrocytes to upregulate the mitochondrial melatonergic pathway, as shown in other cell types [82]. Butyrate, as an HDACi, prevents HDAC potentiating YY1 effects at the promoters of many YY1-induced genes, including in the suppression of the excitatory amino acid (EAAT)2 levels, thereby enhancing glutamatergic excitotoxicity [83]. This indicates that the association of YY1 with astrocyte reactivity may be enhanced when gut microbiome-derived butyrate is suppressed. This is mediated via raised HDAC levels, especially when coupled to an attenuated capacity to induce the mitochondrial melatonergic pathway. (See Figure 2) This is relevant to GBH effects, both directly in astrocytes, as well as via the gut microbiome and gut permeability, with consequences for how astrocytes regulate MNs.

GBH-linked alterations in astrocyte metabolism and function will impact on MNs oxidant regulation, metabolism, and mitochondrial function. Mitochondria are the major producers of cell ROS. Many microRNAs (miRNAs) are ROS regulated, with subsequent consequences for patterned gene inductions [84]. Changes in astrocytes that impact oxidant status and mitochondrial function in MNs will therefore have impacts via ROS/miRNAs/gene patterning. This is important to classical ALS pathophysiology, which is contributed to by miRNAs inducing the truncated BDNF receptor, TrkB-T1, such as miR-4813 and miR-34a, whilst TrkB-T1 is suppressed by miR-185-3p [85]. An increased TrkB-T1/TrkB-FL ratio is an integral aspect of excitotoxicity-induced death across neuronal subtypes [86] and has been long associated with MNs loss in ALS [87]. Alterations in astrocyte function, including as arising from the suppression of the mitochondrial melatonergic pathway, may then have significant consequences for MN’s mitochondria, patterned miRNAs, and patterned gene expression, making MNs more susceptible to challenge. Any suppression of the mitochondrial melatonergic pathway in MNs will make these neurons even more vulnerable to challenge.

As to whether parallels to the interactions of astrocytes and neurons occur between Schwann cells and the NMJ will be important to determine, including any dynamic interactions involving the regulation of the mitochondrial melatonergic pathway. An important, though under-investigated, aspect of the melatonergic pathway is the NAS/melatonin ratio. The NAS/melatonin ratio is increased following the activation of a number of receptors, including the AhR, purinergic P2Y1 receptor, and metabotropic glutamate receptor (mGluR5) [88,89]. NAS is a BDNF mimic via its activation of the TrkB-FL [90], suggesting that the replacement of TrkB-FL with TrkB-T1 in glia, MNs, and muscles can significantly alter the consequences of how the AhR, P2Y1r, and mGluR5 regulate the melatonergic pathway and cellular function. As NAS, like BDNF, activates TrkB-FL, a decrease in the TrkB-FL/TrkB-T1 ratio will have consequences for BDNF and NAS effects, as well as for the receptors that increase NAS production. Clearly, the suppression of the mitochondrial melatonergic pathway, as well as TrkB-T1 upregulation, will negate the important trophic support provided by NAS and BDNF. The suppressed capacity of BDNF (and NAS?) at the TrkB-FL is thought to underpin the early loss of fast MNs and fast twitch muscle fibers in ALS [91]. The dynamic interactions of glia and MNs in determining ROS-miRNA regulation of the TrkB-FL/TrkBT1 ratio will be important to determine in future research, including the relevance of GBH induction of miR-34a, as shown in hippocampal tissue [92], in the upregulation of TrkB-T1 in astrocytes, Schwann cells, NMJ and muscles.

The role of suppression of the melatonergic pathway in response to GBH is given further support by data showing GBH to not only increase astrocyte reactivity but also to suppress cholinergic activity, including the levels of the alpha 7 nicotinic acetylcholine receptor (α7nAChR), as shown in the hippocampus [93]. The α7nAChR is induced by melatonin [94] and is an important immune suppressor, including in the challenged gut, where it mediates the vagal nerve suppression of gut dysbiosis and gut permeability [95]. The vagal nerve suppresses spinal microglia activation during neuroinflammation via α7nAChR upregulation and activation [96], with alterations in the autonomic nervous system and vagal nerve activity long appreciated to be an aspect of ALS pathophysiology [97]. The α7nAChR is also expressed on the mitochondrial membrane, where its function has still to be determined [32], as well as in astrocytes, microglia, and Schwann cells [98]. GBH effects, directly and via melatonergic pathway suppression, will lower α7nAChR levels with possible consequences for astrocyte, microglia, vagal nerve, gut microbiome/permeability, and Schwann cell/NMJ alterations in ALS pathophysiology.

Although astrocytes are the most direct regulators of neuronal, including motor neuronal, survival, oxidant status, energy provision, and neurotransmitter release, it is clear that many factors, including the GBH-influenced gut microbiome/permeability and microglia, can have significant impacts on astrocyte function, and therefore on their regulation of MNs survival. GBH significantly upregulates microglia activation [99]. Both astrocytes and microglia have immune qualities and are reactive cells. However, microglia are the determining influence on brain inflammatory processes, including via their interactions with astrocytes, with some data indicating that an inflammatory microglia phenotype in distinct brain regions may be an aspect of aging-associated changes in the CNS [100].

#### Astrocytes and Microglia Interactions

The astrocyte-regulating factors highlighted above also modulate microglia, including LPS, HMGB1, and the damage-associated molecular pattern (DAMP) factor, hsp70 at TLR4 [101,102,103]; gut microbiome-derived butyrate and HDACi [104]; melatonin production and deactivating effects [77]; ALS susceptibility genes, including mtSOD1 [105], TDP-43 [106], FUS [107] and C9ORF72 repeat [108]; NF-kB and YY1 transcriptional regulation of activation [101,109]; BACE1 and amyloid-β production and clearance [110]; and pro-inflammatory cytokines [101]. Astrocytes and microglia also have their activity regulated over the circadian rhythm, including from pineal gland released melatonin [77]. Such synchronization of changes in astrocytes and microglia by factors relevant to ALS pathophysiology, coupled with the direct interactions of these glia cells, contributes to the complexity of changes occurring in the initiation and progression of ALS symptomatology. As developmental exposure to GBH increases a reactive microglia phenotype [111], it is likely that GBH will have independent effects in microglia that have CNS neuroinflammatory consequences, including for astrocyte-neuronal interactions.

### 2.4. Muscle, NMJ, ALS, and Glyphosate-Based Herbicides

Recent work indicates that ALS may be initiated by alterations in muscles, exemplified by the expression of human mtSOD1 in murine muscle, which leads to fatal ALS-like pathology [112]. These authors propose that MN degeneration arises from a non-autonomous mechanism with MN loss being a selective vulnerability to a form of target-deprivation retrograde neurodegeneration, supported by 40–50% MN loss and mitochondriopathy, as well as loss of NMJ pre- and post-synaptic integrity, coupled to a decrease in nAChR [Martin and Wong, 2022]. Such data lend support to a ‘bottom-up,’ muscle-driven ALS pathoetiology. An important aspect of this muscle-driven ALS pathoetiology is muscle-derived BDNF, which is important to NMJ maintenance, with exercise-induced muscle BDNF release slowing ALS progression in rodent models [113]. Such important effects of muscle-derived BDNF would indicate that melatonergic pathway-derived NAS from different cellular sources, including muscles, satellite cells, Schwann cells, and the NMJ, may be a relevant aspect of NMJ and muscle loss in ALS. This would also suggest that the P2Y1r, AhR, or mGluR5 receptor activation in these different cells, via backward converting melatonin to NAS and subsequent NAS release, may be relevant to NMJ maintenance. The melatonergic pathway, and its regulation by the above receptors, requires investigation across the indicated cells relevant to NMJ-muscle interactions. In the hippocampus, NAS not only activates TrkB but also induces BDNF [114], suggesting that variations in the NAS/melatonin ratio and the mitochondrial melatonergic pathway may be a relevant regulator of muscle BDNF release to the NMJ.

Is there a coordinated decrease in muscle BDNF and NAS with NMJ BDNF, NAS, and/or the melatonergic pathway that combine to underpin MN’s mitochondrial ROS-driven miRNAs in the upregulation of TrkB-T1? Data in other cell types indicates that BDNF increases TPH and, therefore, serotonin, which is a necessary precursor for the melatonergic pathway [115]. As to whether this is relevant to the TPH-serotonin-NAS-melatonin pathway in Schwann cells or the NMJ from muscle-derived BDNF requires investigation. This would indicate that the state of muscle mitochondrial function, as indicated by levels of BDNF release, would be a form of mitochondria-to-mitochondria metabolic communication. Cells, where TPH2 is the predominant isoform in serotonin synthesis, will require 14-3-3e to stabilize TPH2, thereby also implicating factors acting to regulate 14-3-3e availability.

BDNF release from muscle and associated satellite cells to the NMJ is generally regarded as necessary for NMJ maintenance. However, recent work indicates that BDNF is not so much a trophic and/or proliferative factor in peripheral tissues but more of a mitochondrial metabolic regulator [116]. The muscle autocrine and satellite paracrine effects BDNF at the muscle TrkB-FL increases mitogenesis and AMPK to optimize muscle and NMJ mitochondrial function, whilst BDNF at the muscle TrkB-T1 regulates contractility, with pro-BDNF at the p75NTR linked to raised ROS and suboptimal mitochondrial function [116]. An increase in ROS and oxidative stress is associated with raised levels of advanced glycation end (AGE) products, which prevent the cleavage of pro-BDNF to BDNF, thereby enhancing the damaging effects of pro-BDNF at the p75NTR. This would suggest that factors initiating suboptimal mitochondrial function would enhance the production of pro-BDNF at the expense of BDNF. Increased AGE and the activation of its receptor (RAGE) are aspects of ALS pathophysiology, including via effects in astrocytes, microglia, and in gut bacteria [117]. It is also of note that pro-BDNF accumulation increases the inducible cAMP early repressor (ICER) protein, which then occupies the cAMP response element (CRE)-binding sites within the BDNF promoters II and IV, thereby suppressing BDNF induction and associated trophic and plasticity effects [118]. Overall, the regulation of muscle BDNF synthesis and differential effects at different BDNF receptors is intimately related to alterations in mitochondrial function, ROS, and changes in patterned miRNAs and genes expressed.

Terminal Schwann cells are non-myelinating glia present at all human NMJs [119]. Terminal Schwann cells are crucial to the formation and maintenance of the NMJ, as well as the communication between muscles and MNs, and in reinnervation that occurs following injury. Terminal Schwann cells also show significant changes over aging [120]. Over the course of development, MNs release vesicular ATP that activates the purinergic P2Y1r on terminal Schwann cells, which is crucial to the development and maintenance of the NMJ [121]. As to whether the P2Y1r induces the ‘backward’ conversion of melatonin to NAS in terminal Schwann cells, as in the pineal gland [122], leading to NAS- (and NAS-induced BDNF)- TrkB-FL activation at the NMJ, indicating a role for terminal Schwann cell melatonergic pathway in NMJ-muscle maintenance will be important to determine. This will be important to determine in future research, as it would have a number of implications for a diverse array of medical conditions involving NMJ dysregulation. This may also suggest parallels between terminal Schwann cells and astrocytes/enteric glial cells regarding an underestimation of their role within a classical, neuron-centric causal conceptualization of human conditions where neurons are lost.

GBH has significant consequences for the development and function of muscles. Early developmental exposure to GBH results in a number of changes potentially relevant to ALS pathophysiology, including soleus muscle fiber and nuclei reductions, as well as muscle fibrosis and significantly decreased NMJ levels [123]. This is supported by other data showing high GBH levels suppress muscle levels and contraction force in association with alterations in mitochondrial oxygen consumption, elevated free radical levels, and DNA damage arising from mutation accumulation [124]. GBH has significant impacts on the muscle development of a diverse array of aquatic and land-dwelling animals [62], suggestive of impacts on evolutionarily conserved core processes. As to whether GBH effects in muscle are potentiated following GBH exposure during exercise will be important to investigate, including impacts on the mitochondrial melatonergic pathways in muscle, the NMJ, and terminal Schwann cells, as well as in the interactions among these cells.

There is also a growing interest in the role of satellite cells in ALS pathophysiology [125]. Satellite cells are classically conceptualized as embryo-derived stem cells that act as adult skeletal muscle cell precursors, allowing muscle cells to regenerate following damage. An attenuated capacity of satellite cells to regenerate damaged muscle has been proposed to contribute to ALS pathoetiology [126]. Satellite cells are Itga7-expressing cells. Recent data shows another type of Itga7-expressing cell in muscle fiber, referred to by the authors as Itga-7+ glial cells, which, upon muscle nerve lesion, release neurotrophins and tenascin C to contribute to NMJ repair following acute nerve injury [127]. In a preclinical ALS model, these authors showed Itga-7+ glial cells to gradually increase but with an impaired capacity to produce neurotrophins and to repair the NMJ [127]. As with terminal Schwann cells, it will be interesting to investigate the role of the mitochondrial melatonergic pathway in Itga-7+ glial cells in the regulation of their optimized function. As with astrocytes [128], it is highly likely that enteric glial cells and Itga-7+ glial cells express the mitochondrial melatonergic pathway and that variations in that pathway will modulate their function and fluxes. Clearly, this requires future investigation.

### 2.5. Mitochondria, ALS, and Glyphosate-Based Herbicides

A number of studies have highlighted the role of suboptimal mitochondrial function, including as a consequence of ALS risk mutations altering mitophagy [129]. Higher levels of TDP-43 have been found to localize in ALS mitochondria, in association with decreased OXPHOS [130], with suppressed OXPHOS also evident in association with other ALS susceptibility genes [131,132]. Mitochondrial dysfunction is strongly associated with ALS progression, usually coupled to cellular bioenergetics imbalance, dysregulated calcium homeostasis, and alterations in the electron transport chain [133]. As indicated above, such mitochondrial changes are intimately linked to raised ROS levels [134], in turn inducing changes in the patterning of miRNAs and subsequent gene patterning. Consequently, as recognized by many clinical researchers, the optimization of mitochondrial function is a significant treatment target in ALS, as in many other medical conditions. The tethering of the mitochondrial outer membrane to the endoplasmic reticulum membrane is also altered in ALS post-mortems, contributing to Ca2+ and wider mitochondrial and cellular dysregulation [135]. A number of strategies have been investigated as to their clinical utility in optimizing mitochondrial function in preclinical ALS models, including voltage-dependent anion channel (VDAC)1 inhibition [136] and mitophagy inhibition [129].

Most data on mitochondria alterations in ALS have been derived from preclinical models expressing ALS susceptibility genes. In the FUS preclinical model, FUS interacts with DHX30, a mitochondrial RNA granules component needed in mitochondrial ribosome assembly, to decrease its levels. This is also evident in the spinal MNs of ALS patients and is associated with decreases in OXPHOS assembly and function [131]. Alterations in mitochondrial metabolism are also evident in some ALS immune cells, including alterations in TFAM levels and glycolytic metabolism in lymphoblasts [137], as well as in early-stage ALS muscles [138]. The ALS susceptibility gene, profilin 1 mutation, is also associated with suboptimal mitochondrial function [139], highlighting mitochondrial dysfunction as a common feature of ALS susceptibility genes.

As indicated throughout, many of the direct and indirect effects of GBH are mediated via impacts on mitochondrial function in different cells and tissues across the body. The direct and indirect effects of GBH on muscle mitochondrial metabolism have consequences for levels of muscle BDNF and NAS production that optimize mitochondrial function at the NMJ via TrkB-FL activation. Any suppression of the mitochondrial melatonergic pathway in muscles and the NMJ, including from a decrease in 14-3-3 isoforms or acetyl-CoA or TPH2, will attenuate mitochondrial antioxidant availability, thereby increasing ROS and ROS-dependent miRNA. Consequently, there is a change in gene patterning in the NMJ that is at least partly dependent upon the communication derived from alterations in muscle mitochondrial function, where similar ROS-miRNA-gene patterning has occurred. This is a conceptualization of core processes in muscles and the NMJ where an emphasis is placed on the importance of mitochondrial function and the role of the melatonergic pathway. The mitochondrial melatonergic pathway will have local autocrine and paracrine effects as well as impacting on longer-distance communication, in this case between muscles and the NMJ and Schwann cells. The capacity to upregulate the melatonergic pathway thereby endows the cell with resistance to challenge, including oxidant and inflammatory, as is evidenced by the plethora of studies showing the utility of exogenous melatonin in a wide array of different cells under varies types of challenge.

Gut microbiome-derived butyrate is also important in the optimization of mitochondrial function and the restoration of homeostatic processes across an array of challenges, including endometriosis lesions [140]. Butyrate’s optimizing of mitochondrial function seems, at least partly, via its upregulation of mitochondrial sirtuin-3 and PDC disinhibition, thereby increasing pyruvate conversion to acetyl-CoA, which is a necessary cosubstrate for the initiation of the melatonergic pathway. (See Figure 2). This would indicate that the efficacy of gut microbiome-derived butyrate depends on the integrity of the tryptophan-14-3-3e-TPH2-serotonin-acetyl-CoA-14-3-3z-AANAT pathway and associated upregulation of the melatonergic pathway [82]. As indicated, a wide array of factors can limit the availability of the melatonergic pathway, including genetic and epigenetic influences. (See Figure 2). The data indicating increased pTDP-43 in the gut and more systemically, often years before typical ALS symptomatology, as well as pTDP-43 in mitochondria [141], indicate that slow changes act to alter such homeostatic inter-mitochondrial communications over many years in ALS pathoetiology.

Many GBH effects are metabolic, including alterations in the regulation of the mitochondrial TCA cycle [142]. Other data shows alterations in serum metabolites following GBH exposure, including fatty acids metabolites and factors involved in purine biosynthesis [62]. These authors found that most GBH-influenced alterations in serum metabolites were ketogenic amino acids, including leucine, lysine, and tyrosine, which can be oxidized to generate acetyl-CoA for ketone body synthesis [62]. These authors also found methionine and isoleucine to be differentially regulated by GBH. Given that these metabolites are convertible into propionyl-CoA and subsequently to succinyl-CoA, which then enter the TCA cycle [143], this would suggest another route whereby GBH may impact mitochondrial metabolism. Many of the small-effect ALS susceptibility genes have consequences linked to suboptimal mitochondrial function, and the accumulation of such small-effect susceptibility genes has been argued to underpin sporadic ALS [144]. In this sense, GBH exposure would be a mitochondria-modulating environmental risk factor.

GBH can act on an array of processes that influence inter-mitochondrial signaling, including exosomes containing miRNAs that suppress 14-3-3 and the initiation of the mitochondrial melatonergic pathway, such as miR-7, miR-375, miR-451, and miR-709, as well as a number of other epigenetic processes [54]. The influence of GBH on such miRNAs directly in cells and/or via their presence in exosomes requires investigation.

Overall, variations in mitochondrial function are an integral aspect of not only single-cell function but also intercellular communication. GBH impacts mitochondrial function, as well as interacting with a number of surfactants, which potentiates mitochondrial and cellular toxicity [145]. Many of the benefits of an optimized gut microbiome are mediated via butyrate impacts on mitochondrial function across a diverse array of brain and systemic cells. However, butyrate-induced sirtuin-3 and PDC disinhibition leading to increased acetyl-CoA and ATP from the TCA cycle and OXPHOS will be significantly modulated by the capacity of butyrate to upregulate the mitochondrial melatonergic pathway. Consequently, the homeostatic regulatory effects of the gut microbiome-derived butyrate will be significantly altered when the melatonergic pathway is suppressed. Importantly, the capacity of the gut microbiome and butyrate to re-establish homeostasis would be predicted to be significantly compromised when the mitochondrial melatonergic pathway is suppressed in target cells/tissue. As to whether GBH is a significant regulator of such intracellular, local, and inter-area mitochondrial communication will be important to determine in future research, including how GBH interacts with the small effect ALS susceptibility genes in the pathogenesis of sporadic ALS.

## 3. Integrating ALS Pathophysiology

There is a growing appreciation that many conditions that have been pathophysiologically defined by mutations may be more appropriately seen as disorders in mitochondrial metabolism, driven by mitochondrial ROS-mediated DNA damage, including in cancers [146,147], autism spectrum disorders [148], myalgic encephalomyelitis/chronic fatigue syndrome [149], multiple sclerosis [14] respiratory disorders [150] and many neuropsychiatric conditions [151]. As indicated above, ALS can similarly be defined as arising from alterations in mitochondrial function and the consequences that this has for intercellular processes. The failure to restore intercellular homeostasis arises from dysregulated mitochondrial ROS driving alterations in patterned miRNAs and, consequently, in patterned gene expressions and cellular fluxes. The ability to upregulate the mitochondrial melatonergic pathway is an important aspect of these intracellular and intercellular alterations, providing a novel conceptualization of ALS pathoetiology and pathophysiology. It is proposed that alterations in mitochondrial function occur at three key hubs in ALS, namely the NMJ, glia, and muscle. These key hubs are importantly regulated by systemic processes, namely the gut microbiome/permeability and circadian rhythm, providing a pathophysiological frame of reference upon which it is proposed that GBH can act to increase ALS susceptibility.

This simple outline is complicated by alterations in immune cell function by changes in the gut microbiome/permeability [13] and circadian rhythm [152], as well as the array of altered fluxes arising from glia, muscle, and MNs/NMJ, mediated by changes in their mitochondria-ROS-patterned miRNAs. This has treatment and preventative implications, as indicated below (Treatment Implications Section}, including the targeting of the mitochondrial melatonergic pathway in key cellular hubs, thereby allowing challenged cells to return closer to their previous homeostatic interactions with other cells. Prevention and treatment derived from such a perspective requires a focus on core aspects of physiological function rather than the plethora of differentially regulated genes and protein effluxes that are currently mooted as treatment targets in ALS and other similarly complex and poorly treated metabolic-systemic medical conditions. Some key aspects are highlighted next (See Figure 2).

BDNF regulation in ALS pathophysiology may be intimately linked to alterations in the melatonergic pathway. As noted, a number of miRNAs can regulate the melatonergic pathway, including miR-7, miR-375, miR-451, and miR-709. Some of these miRNAs are also significant regulators of BDNF [153,154], suggesting that the important role of BDNF in ALS, especially muscle released BDNF at the NMJ, may be intimately linked to alterations in the mitochondrial melatonergic pathway. Clearly, this will be important to be determined by future research, including the factors that may be differentially regulating these miRNAs across different cell types in ALS patients.

ALS is often diagnosed in late middle age, suggesting an aging-associated factor. As noted, astrocytes are an important hub in ALS pathophysiology. Rodent astrocytes that have been cultured from old (26–29 months) vs. young (4–6 months) mice show a number of significant changes, including slower growth rate, attenuated mitochondrial membrane potential, heightened sensitivity to oxidative challenge, and an enhanced capacity of ATP to induce high Ca2+ responses, which may arise as a consequence of suppressed mitochondrial Ca2+ sequestration [155]. The co-culturing of neurons with aged astrocytes leads to neurons with a heightened sensitivity to oxidative challenge, as to whether these aging-associated changes in astrocytes and their capacity to afford protection to neurons are mediated by a decrease in the astrocyte melatonergic pathway requires investigation. Should the tenfold decrease in pineal melatonin release over human aging (from 18 yrs to 80 yrs) be replicated in astrocytes, this would have significant implications for astrocyte regulation of neuronal function and survival, with clear relevance to how aging associates with neurodegenerative conditions, such as Alzheimer’s disease and Parkinson’s disease as well as ALS.

As noted, the melatonergic pathway is an important determinant of the activation/deactivation of both central and systemic immune cells [77,78]. As melatonin increases sirtuins, leading to more optimized mitochondrial function, including the suppression of astrocyte PDK [156], any aging-associated loss of astrocyte melatonin production, release, and autocrine/paracrine effects may be a significant mediator of aging susceptibility to neurodegenerative conditions via impacts on neuronal, including MNs, mitochondrial function. Data showing a 1.5–4.9-fold reduction in hippocampal and saliva sirtuin-1, sirtuin-3, and sirtuin-6 in dementia patients vs. controls [157] may be intimately linked to the suppression of both pineal and local melatonin production. As PDK inhibits PDC, leading to a decreased conversion of pyruvate to acetyl-CoA in astrocytes and a consequent increase in lactate production and release, increased PDK will result in lower levels of astrocyte OXPHOS, coupled with raised levels of lactate production and release [156]. PDK and enhanced glycolysis are crucial for the activation of almost all immune cells due to the heightened metabolic requirements over the course of activation [158], suggesting that aging-associated suppression of the melatonergic pathway in reactive cells, including astrocytes and microglia, may underpin the heightened low-level inflammation often evident in the pathoetiology of neurodegenerative disorders, including ALS.

Melatonin production from the melatonergic pathway would be further suppressed over aging by the enhanced AhR levels and ligands in the course of aging [159], given that the AhR not only backward converts melatonin to NAS but may also act to suppress 14-3-3 and, therefore the stability of the first enzyme in the melatonergic pathway, AANAT [33]. As noted, the suppression of NAS, especially at the NMJ, may be of particular importance in ALS, given the significant role that TrkB-FL activation by BDNF or NAS has on the maintenance of the NMJ, thereby indicating that a decrease in tryptophan availability or uptake, and its conversion by 14-3-3e-stabilized TPH2, may be of particular importance in ALS, especially at the muscle-NMJ interface, with any factors suppressing the availability of NAS likely to increase MNs susceptibility to apoptosis.

Notably, alterations in astrocyte redox status can dramatically alter ionic regulation [160]. These authors showed that astrocyte redox status, primarily GSH regulation, dramatically altered Ca2+ responses to Gq-linked purinergic P2Yr activation, with impacts on IP3-mediated store-operated Ca2+ entry, thereby modulating astrocyte excitability under purinergic stimulation. The inhibition of GSH synthesis in astrocytes and microglia has long been recognized to enhance neuroinflammatory responses [161]. Activation of nuclear factor erythroid-derived 2-related factor 2 (Nrf2), through binding to antioxidant response elements (AREs), inhibits the induction of pro-inflammatory factors, including BACE1 and, therefore, amyloid-β production [162]. This will have consequences for the amyloid-β accumulation around dying MNs. Notably, melatonin activates the Keap1-Nrf2-ARE-GSH pathway in astrocytes [163], indicating that the suppression of pineal melatonin or astrocyte-derived autocrine melatonin, will enhance inflammatory responses to challenge, coupled with alterations in ionic and metabolic regulation. As such, the inhibition of the tryptophan-serotonin-melatonergic pathway by GBH suppression of the shikimate pathway, decreased tryptophan and serotonin availability following gut dysbiosis and increased gut permeability, coupled to 14-3-3 and TPH2 suppression, as well as aging-associated decrements in melatonergic pathway induction, will all contribute to suppressing core aspects of astrocytic regulation of neuronal function and survival. Systemic TNF-α can suppress pineal melatonin production [164], implicating wider systemic inflammation in the melatonin-mediated homeostatic circadian and local regulation of mitochondrial function in immune cells, including glia [14].

Such processes, as exemplified by astrocytes, are also important at other ALS hubs, including the muscle-NMJ interface and the interactions of the gut microbiome/permeability with intestinal epithelial cells, enteric glial cells, enteric nervous system, and vagal inputs. More specific to astrocytes, the effects of an increased LPS/HMGB1-TLR4/YY1 pathway results in the YY1 suppression of EAAT2 in astrocytes [165], leading to upregulated glutamatergic activity and associated excitotoxicity in MNs, which is the rationale for the use of Riluzole in ALS treatment. This is an important aspect of ALS pathophysiology and may be compensated by an increase in mGluR5 activation, which can ‘backward’ convert melatonin to NAS, thereby having trophic effects at the BDNF receptor, TrkB-FL. However, such compensatory effects of upregulated glutamatergic input onto MNs would be suppressed in the presence of an increased TrkB-T1/TrkB-FL ratio. As such, the rationale for current pharmaceutical treatments in ALS, namely Riluzole and Edaravone, would have their putative therapeutic efficacy (glutamatergic and antioxidant, respectively) intimately linked to the regulation of the mitochondrial melatonergic pathway. As many YY1 transcriptional effects are potentiated by HDAC, the suppression of gut microbiome-derived butyrate and its capacity as an HDACi would be expected to enhance YY1 transcriptions. This again highlights the importance of alterations in the gut microbiome in shaping astrocytic regulation of MNs in the course of ALS pathophysiology.

### GBH and Integrated ALS Pathophysiology

There is a growing appreciation that GBH may be associated with a wide array of diverse medical conditions, including cancer (especially non-Hodgkin’s lymphoma), neurodegenerative conditions, and neuropsychiatric conditions. As indicated, all of these conditions can be conceptualized as disorders in mitochondrial metabolism [14,146,147,148,149,150,151]. The preclinical data showing GBH exposure prenatally and early postnatally to have impacts on muscles and the NMJ [123], as well as gut microbiome composition and levels of short-chain fatty acids [166], could indicate a role for GBH in the pathoetiology of host of diverse medical conditions, including autism [166] and ALS [167].

As well as data showing GBH to impact an array of different cells and important hubs in ALS pathophysiology, GBH also has significant effects on circulating platelets in preclinical [168] and human investigations [169], which can contribute to alterations in the BBB [14] and gut microbiome/permeability [170], as well as muscle homeostasis [171] and muscle-atrophy associated sarcopenia [172]. TDP-43 levels are significantly increased in ALS platelets, in correlation with symptom severity [173], whilst the gradual loss of mitochondrial SOD2 in platelets over aging is proposed as a significant factor in an array of aging-associated conditions [174]. Platelet-rich plasma is important in the clinical management of muscle repair, including in professional sports [175]. It will be important for future research to determine whether GBH regulates mitochondrial platelet function and the mitochondrial melatonergic pathway, including via alterations in mitochondrial ROS-induced regulation of 14-3-3-regulating miRNAs, as indicated by data in autism [176]. Some consequences of GBH effects on core hubs in ALS are shown in Table 1.

Data in rodents indicate that early developmental exposure to GBH alters ROS and endogenous antioxidants as well as decreasing mitochondrial SOD2, indicative of enhanced mitochondrial ROS and associated with the patterning of miRNAs and genes induced. Early developmental GBH exposure also has consequences in adulthood in preclinical models, leading to elevations in oxidants (malondialdehyde) and endogenous antioxidant enzymes (catalase) coupled with a decrease in neurogenesis, suggestive of maintained oxidative metabolic challenge [177]. Given the data indicating the detrimental impacts of early developmental exposure to GBH on muscle fiber, nuclei, and fibrosis, as well as on suppressed NMJ levels [123], this is suggestive of early developmental priming that shapes the homeostatic nature achieved, as influenced by alterations in mitochondrial function and the impact this has on inter-cellular communication. As highlighted throughout, the role of the mitochondrial melatonergic pathway and its regulation by environmental factors in development and adulthood, including by GBH, in shaping such altered homeostatic processes will be important to determine.

This frame of reference allows the integration of wider bodies of previously disparate data on ALS, including occupational risk and the growing number of studies highlighting the relevance of variations in mitochondrial function and metabolism across a range of diverse CNS and systemic cells. The emphasis on the role of the mitochondrial melatonergic pathway also allows for the incorporation of recent data highlighting the pathophysiological role of chitinase-3-like protein 1 (CHI3L1), also known as YKL-40, in ALS pathophysiology [178] as well as in an array of other medical conditions, including neurodegenerative disorders more widely [179], asthma [180], autoimmune disease [181] and cardiovascular disorders [182].

YKL-40 is highly expressed in reactivated astrocytes, where it is proposed to mediate the effects of hyperphosphorylated tau in Alzheimer’s disease preclinical models [183]. In glioma, YKL-40 binds NFKB1 to upregulate NF-kB signaling [184], whilst in epithelial cells, NF-kB suppresses miR-149-5p to upregulate YKL-40 following TLR2/3 activation and induction of TNF-α [185]. This requires investigation in glial cells but would suggest that the incapacity of NF-kB to upregulate the melatonergic pathway may allow for interactions of YKL-40 and NF-kB in a positive feedback loop that maintains inflammation when the anti-inflammatory signaling of released melatonin is absent. As GBH increases NF-kB across a range of species and cell types [186] and potentiates LPS in upregulating TLR2 [187], GBH may enhance this putative NF-kB/YKL-40 positive feedback loop when the melatonergic pathway is suppressed. This would also have consequences for GBH effects in the gut, and gut-associated regulation of CNS processes, with GBH upregulating circulating LPS and TLR4 activation in astrocytes, leading to NF-kB and YY1 upregulation, with NF-kB effects potentiated by YKL-40 induction in the absence of any intracrine or autocrine melatonin induced by NF-kB. Endogenous TLR4 activators, such as HMGB1 and hsp70, would also be predicted to have significant additive/synergistic interactions with GBH. Such changes would not be expected to be restricted to glia and ALS but evident across neurodegenerative conditions and cancers [184]. This highlights how alterations in the regulation of core processes underpinning mitochondrial function may integrate previously disparate bodies of data on ALS pathophysiology.

How YKL-40 interacts with variations in gut microbiome-derived butyrate in the regulation of NF-kB will be interesting to determine. NF-kB upregulates YKL-40 in glioma cells [188], suggesting that the impacts of variations in the gut microbiome and gut permeability may significantly upregulate NF-kB in CNS cells, which would be potentiated by GBH. In glioblastoma, HDAC1 suppresses NF-kB induction of YKL-40, suggesting that gut microbiome-derived butyrate may potentiate the NF-kB upregulation of YKL-40. Such possible contrasting effects of butyrate will be important to determine in reactive astrocytes and microglia, as any butyrate-mediated upregulation of NF-kB-YKL-40 would suggest gut microbiome-derived butyrate may then contribute to maintained low-level inflammation, as is often thought to underpin most neurodegenerative conditions, in circumstances where the mitochondrial melatonergic pathway is suppressed. As to whether the suppression of glia mitochondria melatonergic pathway parallels infection-driven TNF-α suppression of pineal melatonin production [164] will be interesting to determine, including how this varies over time. Pineal melatonin suppression is proposed to allow local resolution of inflammation, which is then homeostatically reinforced by subsequent pineal melatonin release. Interestingly, circulating pro-inflammatory cytokines also increase gut permeability and associated gut dysbiosis, indicating that CNS and systemic inflammation may co-ordinate pineal melatonin suppression with gut permeability/dysbiosis, allowing the resolution of local inflammation to re-establish the gut barrier, microbiome, and circadian melatonin, which would then reinforce the homeostasis attained following inflammation. Such delayed responses of the gut and pineal gland do not always seem beneficial, as exemplified in preclinical spinal lesion studies [189], where the use of melatonin or butyrate aids recovery, although both are suppressed by the immune cell pro-inflammatory responses to injury. To what extent parallels occur during the course of CNS inflammation as a consequence of microbial type signaling via TLR2/4 will be important to determine. Further research questions posed by the complexity of YKL-40 effects and interactions are indicated in the Future Research Section below.

## 4. Future Research

Do the specific enzymes involved in muscle mitochondrial metabolism [190] make the muscle more susceptible to the direct and indirect effects of GBH? Would specific GBH-susceptibility aspects of muscle mitochondrial metabolism be more evident in ‘fast twitch’ muscle and/or in associated satellite cells and their interactions with muscle? How relevant are variations in the mitochondrial melatonergic pathway in muscle and/or satellite cells to this?Given that many of butyrate’s beneficial effects may be mediated by its capacity to upregulate the mitochondrial melatonergic pathway, does butyrate regulate the conversion of tryptophan to melatonin, and therefore the mitochondrial melatonin availability, across different cell types, including astrocytes. i.e., Does butyrate modulate 14-3-3e, TPH2, 14-3-3z, acetyl-CoA, AANAT, ASMT across different cell types?Melatonin prevents the toxic effects of GBH in neurons, testes, oocytes, pregnancy, liver, and kidney [191,192,193]. Do such wide-ranging effects suggest that GBH decreases the mitochondrial melatonergic pathway, as perhaps the detrimental effects of GBH on mitochondrial function could suggest?Do the alterations in glia regulation of MNs involve elevations in MNs mitochondria ROS, leading to raised levels of miR-4813 and miR-34a, thereby increasing TrkB-T1, as in hippocampal neurons [92], underpinning the loss of NAS and BDNF trophic support? Is the loss of muscle derived BDNF at the NMJ of particular importance in ALS pathoetiology?Does GBM interact with specific aspects of muscle metabolism that become potentiated when being physically active in GBM-containing environments, contributing to the occupational risk of ALS [48,49,50,51]?Does the regulation of the NAS/melatonin ratio in muscles, Schwann cells, and NMJ determine key aspects of NMJ maintenance, including the release of muscle BDNF to TrkB at the NMJ?Is aging associated with a significant decrease in the capacity of astrocytes, microglia, Schwann cells, and MNs to induce and produce melatonin? Does low-level inflammation contribute to aging associated melatonin suppression? Do pro-inflammatory cytokines co-ordinate gut permeability/dysbiosis with suppressed pineal melatonin production?Does microglia activation suppress the capacity of astrocytes to induce the melatonergic pathway, paralleling the effects of pro-inflammatory cytokine suppression of pineal melatonin [164]? Would this suggest a ‘dominance’ type effect of microglia in the regulation of inflammatory astrocyte-neuronal interactions via impacts on the capacity of astrocytes and neurons to upregulate the melatonergic pathway, allowing these immune cells to be ‘in charge’? Would this parallel the putative effects of cancer cells and cancer stem-like cells in the tumor microenvironment, where they act to regulate metabolism in neighboring cells [146].Does the maintenance of low-level inflammation maintain an increase in gut permeability in conjunction with gut dysbiosis and suppression of short-chain fatty acids, including butyrate, thereby maintaining LPS/TLR4/NF-kB/YY1 and astrocyte activation?Does the mitochondrial melatonergic pathway and its regulation by environmental factors in development and adulthood, including by GBH, shape the nature of homeostatic processes as determined by variations in mitochondrial function in different cell/tissue microenvironments?To what extent are the miRNAs that regulate 14-3-3 and the melatonergic pathway coordinating BDNF release [153,154], especially in muscle-derived BDNF at the NMJ?How do the astrocyte-like enteric glial cells interact with gut microbiome products and vagal inputs, including any direct and indirect GBH effects?Do enteric glial cells and Itga-7+ glial cells express the mitochondrial melatonergic pathway, and is this altered in ALS, including by GBH, leading to alterations in how these astrocyte-like cells form crucial hubs to regulate important cellular and intercellular changes within the gut and mucosal immune system in ALS pathoetiology?Are Itga-7+ glial cells generated from satellite cells under challenge at the NMJ-muscle interface?Does GBH directly or indirectly (e.g., via the gut microbiome) impact the capacity of Itga-7+ glial to be generated/proliferated?How relevant is the inhibition of the Shikimate pathway in the ALS gut microbiome, including by GBH? Is this mediated via GBH effects on Akkermansia muciniphila?As well as providing tryptophan for the synthesis of serotonin, NAS and melatonin, the Shikimate pathway also provides tryptophan to be converted to tryptamine in the gut by tryptophan decarboxylase, allowing tryptamine to activate the AhR, thereby sealing the gut barrier [194]. Is the shikimate pathway induction of tryptophan for tryptamine activation of the gut AhR coordinated with an increase in gut microbiome-derived butyrate, as some data indicates [194]?Does the activation of the purinergic P2Y1r, glutamatergic mGluR5, and/or AhR regulate NAS availability in astrocytes, Schwann cells, muscles, and/or the NMJ?Is there a coordinated decrease in NAS and/or the melatonergic pathway in MNs that is coordinated with consequent mitochondrial ROS and ROS-induced miRNAs upregulating TrkB-T1?Does muscle-derived BDNF upregulate the TPH-serotonin-NAS-melatonin pathway in Schwann cells or the NMJ?How relevant are alterations in platelet mitochondrial melatonergic pathway in ALS pathophysiology, especially in muscle regulation? Does GBH have effects on platelets that contribute to ALS pathoetiology?Does GBH alter levels of the short-chain fatty acids butyrate, propionate, and acetate in the human gut microbiome?Preclinical data indicate that GBH increases BBB and gut permeability [64,195]. Relevance during exposure and after GBH exposure in humans?There is a growing appreciation of an early developmental origin to an array of diverse medical conditions. Would this be relevant in ALS, including from GBH exposure?Does GBH regulate mitochondrial function and the mitochondrial melatonergic pathway, thereby changing inter-area mitochondria/cellular communication, underpinning the systemic changes occurring over the course of ALS pathoetiology and pathophysiology? Does GBH impact mitochondrial ROS-regulated miRNAs involved in 14-3-3 isoform suppression?GBH can act on an array of processes that influence inter-mitochondrial signaling, including exosomes containing miRNAs that suppress 14-3-3 and the initiation of the mitochondrial melatonergic pathway, such as miR-7, miR-375, miR-451, and miR-709. The influence of GBH on such miRNAs directly in cells and/or via their presence in exosomes requires investigation.Given the heightened levels of NK cells and CD8^+^ t cells in ALS patients [20], in association with elevations in IFN-γ, TNF-α, I, and IL-17a [20], is this indicative of increased IL-17 producing, as well as classical IFN-γ producing, NK cells and/or γδ17 t cells or Th17 cells?γδ17 t cells are significant regulators of muscle response to challenge [196] but have not been investigated in ALS pathophysiology. Is there a role for alterations in the levels and/or ratio of IFN-γ or IL-17 producing γδ t cells in ALS? Does GBH have any impact on γδ17 t cells or γδ-IFN-γ t cell levels and/or ratio? Would this be relevant to GBH effects in muscles? GBH is significantly linked to non-Hodgkin’s lymphoma in many studies [197], which includes primary cutaneous γδ T-cell lymphoma (PCGDTL) as a rare subtype [198]. γδ t cell and NK cell activity are also significant determinants of treatment response in B-cell non-Hodgkin’s lymphomas [199].YKL-40 effects are further complicated by the YKL-40 inhibition of Receptor for advanced glycation end products (RAGE) activation by S100A9 [200], which may be more relevant in Alzheimer’s disease than ALS [201]. However, the possible regulation of RAGE activation by YKL-40 raises a number of questions. Does the increase in amyloid-β in ALS that activates RAGE contribute to the elevated RAGE signaling in the progression of ALS [202]? Would YKL-40 modulate such a putative NF-kB/YY1-induced BACE1/ amyloid-β activation of RAGE, as well as other RAGE ligands, including HMGB1 and other S100/calgranulin family members, during RAGE activation?Is there a positive feedback loop in ALS glia for maintained inflammation involving the interactions of YKL-40 and NF-kB when coupled to melatonergic pathway suppression? Is this linked to amyloid-β upregulation across different medical conditions, including glioblastoma, breast cancer, ALS, and Alzheimer’s disease?Does YKL-40 have any role in the suppression of the tryptophan-melatonergic pathway?The Maillard reaction occurs during the interaction of a protein’s free amine groups with the carbonyl groups in carbohydrates, leading to an increase in AGE products and acrylamide, which can be inhibited by tryptophan [203]. Does GBH, via the inhibition of the gut microbiome Shikimate pathway, thereby inhibiting gut tryptophan production, upregulate AGE, RAGE activation, and pro-BDNF levels?Are the mitochondrial alterations evident in ALS associated with changes in the levels of the peptide transporters, PEPT1/2, in the mitochondrial membrane, thereby impacting the levels of exogenous melatonin uptake into mitochondria via PEPT1/2 and the sulfation metabolites of melatonin via the organic anion transporter (OAT)3, as shown in cancer cells [204]?Does gut microbiome-derived butyrate regulate PEPT1? Human PEPT1 is upregulated by the transcription factor, Sp1, but can also be regulated by the caudal-related homeobox transcription factor, Cdx2, in the presence of Sp1 and the gut microbiome-derived butyrate [205]. There is also a circadian regulation of PEPT1, which seems dependent on the CLOCK-controlled protein albumin D, with higher PEPT1 uptake at night, as shown in rodents [206]. Does gut microbiome-derived butyrate ‘reset’ mitochondrial function at night by enhancing pineal-derived melatonin uptake into mitochondria, in turn upregulating the mitochondrial melatonergic pathway? This would indicate an interaction of the gut microbiome and circadian rhythm in the homeostatic regulation of mitochondrial function, as determined by circadian and local melatonin uptake via mitochondrial PEPT1.PEPT1 is down-regulated by miR-193a-3p, whilst PEPT1 activity can regulate a number of miRNAs, as shown in the colon [207]. How does mitochondrial PEPT1/2 interact with other cells more directly relevant to ALS pathophysiology, including glia?

## 5. Treatment Implications

Given the recent data showing significant clinical improvement in ALS patients with gut bacteria-targeted treatment [57] and the relevant effects of gut microbiome-derived products, such as butyrate, on wider ALS pathophysiology, the gut microbiome is a wide-acting treatment target in ALS.Data on GBH-induced changes in the gut microbiome of Eriocheir sinensis show that the addition of melatonin prevents GBH-driven changes [208]. This would suggest the preventative effects of melatonin for people regularly exposed to GBH.Mutations in the ALS risk genes SOD1 and *TARDBP* (TDP-43) in preclinical models have many of their effects on mitochondrial and wider cellular function negated by mitophagy inhibitors [129], suggesting some utility of mitophagy inhibitors in ALS. This requires further clinical investigation.Melatonin has utility in the treatment of ALS [28], as may tryptophan (as a precursor for the serotonergic-melatonergic pathway), whilst sodium butyrate would also be expected to have beneficial effects, although requiring investigation [42]. The benefits of other nutriceuticals in ALS, such as epigallocatechin gallate, resveratrol, and other polyphenols, are also likely to be small in effect.More targeted treatments based on the role of the mitochondrial melatonergic pathway are viable for development. For example, utilizing stem cell exosomes targeted to optimize the tryptophan-melatonergic pathway steps in cells is viable, including via exosomal miRNAs and 14-3-3 isoforms. In early-stage ALS, this could have significant benefits if applied to muscles to optimize muscle NAS and BDNF release. Although requiring investigation, such a treatment procedure is likely to benefit from the concomitant utilization of circadian melatonin and butyrate to dampen the immune response and help to reset the homeostatic interactions of muscles, satellite cells, and the NMJ. The capacity to target such exosomes to an identified marker on a particular cell phenotype with target cell-relevant cargo (such as 14-3-3e) would refine treatment and allow for a shotgun treatment of different target cells and their processes.

## 6. Conclusions

This article reviews and links a wide range of previously disparate data pertaining to ALS pathoetiology and pathophysiology, highlighting the importance of alterations in the regulation of the mitochondrial melatonergic pathway. This pathway seems conserved from the very beginnings of multi-cellular life on earth, when the first ancient bacteria crept into a single-cell organism, gradually evolving into mitochondria. The ubiquitous presence of the melatonergic pathway in all cells (so far investigated), and the equally seemingly ubiquitous benefits of melatonin in animals and plants under challenge, strongly indicate the importance of the melatonergic pathway. Factors dysregulating this pathway are major drivers of complex and often poorly treated human medical conditions, including cancers. The suppression of the mitochondrial melatonergic pathway in ALS not only dysregulates cellular function but alters the factors maintaining the homeostatic regulation of inter-cellular and inter-area function and communication. It requires investigation as to whether GBH exposure contributes to the pathoetiology and pathophysiology of ALS. The investigation of which should better clarify the pathophysiological underpinnings of sporadic ALS and hopefully provide treatments and preventions that are targeted to appropriate biological underpinnings.

## Figures and Tables

**Figure 1 ijms-24-00587-f001:**
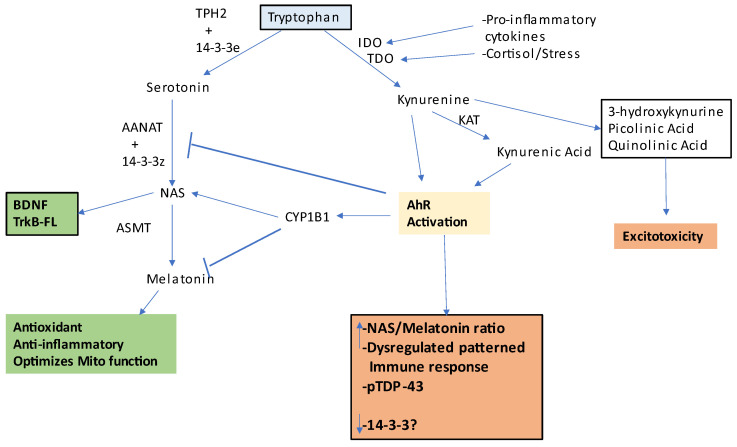
This figure shows how tryptophan (light blue shade) can be utilized for beneficial (green) or detrimental (orange) effects in ALS. Tryptophan is converted to serotonin in astrocytes by tryptophan hydroxylase (TPH)2 which has been stabilized by 14-3-3e. Serotonin is converted by AANAT to N-acetylserotonin (NAS) when stabilized by another 14-3-3 isoform and in the presence of acetyl-CoA as a co-substrate. NAS is converted by ASMT to melatonin. NAS can activate the BDNF receptor, TrkB, as well as induce BDNF, whilst melatonin has antioxidant, anti-inflammatory, and mitochondria-optimizing effects. However, in the presence of pro-inflammatory cytokines and stress-associated cortisol-induced IDO and TDO, respectively, tryptophan is converted to kynurenine, which can activate the AhR leading to a number of detrimental effects via kynurenine pathway products that drive excitotoxicity. However, when an active melatonergic pathway is present, AhR-induced CYP1B1 can ‘backward’ convert melatonin to NAS, thereby increasing TrkB-FL activation and BDNF induction. Abbreviations: AANAT: aralkylamine N-acetyltransferase; AhR: aryl hydrocarbon receptor; ASMT: N-acetylserotonin O-methyltransferase; CYP: cytochrome P450; IDO: indoleamine 2,3-dioxygenase; KAT: kynurenine aminotransferase; NAS: N-acetylserotonin; TDO: tryptophan 2,3-dioxygenase; TPH2: tryptophan hydroxylase.

**Figure 2 ijms-24-00587-f002:**
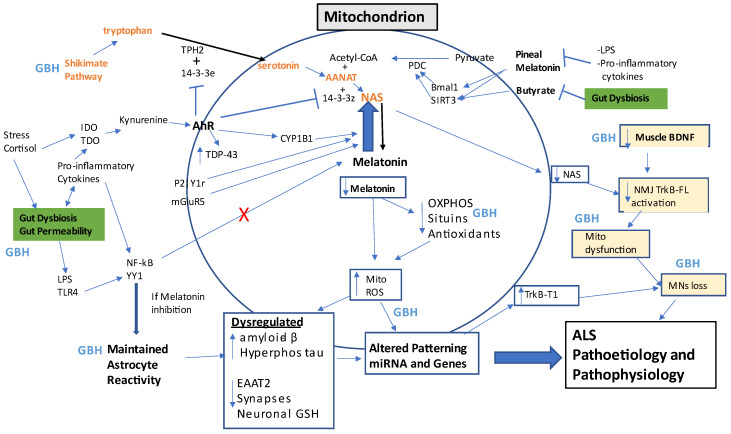
This summary figure shows how many ALS-linked factors can impact mitochondrial function, with relevance across different cell types. Gut dysbiosis is highlighted in green, whilst muscle-NMJ-MNs loss is highlighted in yellow. Gut permeability-associated LPS activates TLR4/NF-kB/YY1 to drive maintained astrocyte activation in the presence of a suppressed melatonergic pathway. The shikimate pathway produces tryptophan (as well as tyrosine and phenylalanine) to drive the serotonin/NAS/melatonin pathway, which requires different 14-3-3 isoforms and acetyl-CoA to be functional. A number of factors can ‘backward’ convert melatonin to NAS, including P2Y1r, mGluR5, and AhR activation. NAS can be beneficial in ALS as it is a BDNF inducer and activator to TrkB-FL. The AhR may also have detrimental effects via 14-3-3 suppression, as well as increasing TDP-43. Gut dysbiosis and pro-inflammatory cytokine suppression of pineal melatonin suppress the capacity of butyrate and melatonin to disinhibit PDC and the PDC conversion of pyruvate to acetyl-CoA, which is a necessary co-substrate of AANAT in the conversion of NAS to melatonin. The suppression of the melatonergic pathway leads to increased oxidants, decreased endogenous antioxidants, OXPHOS, and sirtuins, thereby increasing mitochondrial ROS, ROS-inducing miRNAs, and altered gene patterning, which is a major driver of ALS pathophysiology. The maintained activation of astrocytes decreases neuronal GSH, synapses, and EAAT2 whilst increasing amyloid-β and hyperphosphorylated tau, all of which contribute to ROS-driven changes in patterned miRNAs and gene inductions. The loss of muscle BDNF leads to suboptimal mitochondrial function at the NMJ and MNs loss. All these changes comprise the biological underpinnings of ALS. GBH can have multiple negative impacts, as indicated by points across the figure. Abbreviations: AANAT: aralkylamine N-acetyltransferase; AhR: aryl hydrocarbon receptor; ASMT: N-acetylserotonin O-methyltransferase; BDNF: brain-derived neurotrophic factor; Bmal1: basic helix-loop-helix ARNT Like 1; CYP: cytochrome P450; EAAT: excitatory amino acid transporter; GBH: glyphosate-based herbicides; IDO: indoleamine 2,3-dioxygenase; LPS: lipopolysaccharide; mGluR5: metabotropic glutamate receptor 5; mito: mitochondria; MNs: motor neurons; NAS: N-acetylserotonin; NF-kB: nuclear factor kappa-light-chain-enhancer of activated B cells; NMJ: neuromuscular junction; OXPHOS: oxidative phosphorylation; P2Y1r: purinergic 2Y1 receptor; PDC: pyruvate dehydrogenase complex; ROS: reactive oxygen species; SIRT3: sirtuin-3; TDO: tryptophan 2,3-dioxygenase; TDP-43: transactive response DNA binding protein 43kDa; TLR: toll-like receptor; TPH2: tryptophan hydroxylase; TrkB-FL: tyrosine receptor kinase B-full length; TrkB-T1: tyrosine receptor kinase B-truncated; YYI: yin yang 1.

**Table 1 ijms-24-00587-t001:** Summarizes GBH effects in glia, gut, muscles, and NMJ, with associated references. Abbreviations: 5-HT: 5-hydroxytryptamine (serotonin); α7nAChR: alpha 7 nicotinic acetylcholine receptor; BDNF: brain-derived neurotrophic factor; EAAT: excitatory amino acid transporter; GBH: glyphosate-based herbicides; HDACi: histone deacetylase inhibitor; LPS: lipopolysaccharide; MNs: motor neurons; NAS: N-acetylserotonin; NF-kB: nuclear factor kappa-light-chain-enhancer of activated B cells; NMJ: neuromuscular junction; TPH2: tryptophan hydroxylase; TrkB-FL: tyrosine receptor kinase B-full length; TrkB-T1: tyrosine receptor kinase B-truncated.

GBH Effect	Glia	Gut	Muscles	NMJ	References
Shikimate pathway inhibition;	Decrease tryptophan, 5-HT, NAS, melatonin?	Altered gut bacteria patterning;	Altered 5-HT muscle regulation?	Altered MNs and NMJ 5-HT?	[60,61]
increased gut permeability, circulating LPS	Reactivation, Inflammation, Increased YY1, lower EAATs	Altered gut bacteria interactions with mucosal immune system and enteric glia.	Muscle NF-kB driven inflammation	MNs apoptotic processes	[62,64]
GBH resistant gut bacteria	Lower butyrate/HDACi-linked epigenetic regulation. Altered fungal composition	Gut dysbiosis, HDACi	Lower butyrate HDACi-linked epigenetic regulation. Mitochondrial dysfunction.	Lower butyrate, HDACi-linked epigenetic regulation.	[62,63]
Increased TNF-α and amyloid-β	Reactivation, Inflammation,	Gut permeability	Muscle damage Lower muscle BDNF	MNs loss	[73,75,99,111,123,124]
Increase miR-34a, other miRNAs?	Lower TrkB-FL/TrkBT1 ratio	Gut permeability	Suboptimal muscle function	Lower TrkB-FL/TrkBT1 ratio	[92,123,124]
Suppresses α7nAChR	Lost attenuation of glia and immune reactivity	Gut permeability	Heightened muscle inflammation	Heightened MNs and NMJ inflammation	[93,95,98]
Platelet dysregulation	BBB permeability	Gut permeability	Muscle atrophy	?	[14,170,171,172,173,174,175]

## Data Availability

Not applicable.

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
