# Peer review of "Amyotrophic Lateral Sclerosis Pathoetiology and Pathophysiology: Roles of Astrocytes, Gut Microbiome, and Muscle Interactions via the Mitochondrial Melatonergic Pathway, with Disruption by Glyphosate-Based Herbicides"

_ijms, 2022, doi:10.3390/ijms24010587_

Round 1

Reviewer 1 Report

I read this review by George Anderson with great interest.  It deals the susceptibility to sporadic ALS determined by a dysregulation of the melatoninegic mitochondrial pathway that the Author hypothesizes may be caused by the ingestion of glyphosate-based herbicides with the diet. An important characterizing point of this hypothesis is to identify a fundamental role in the intestine and in particular in the altered intestinal microbiota. This is not a "neutral" review on the subject but is a real thesis that the author pursues with a careful and complete collection of all the literature in support of it. Despite the large number of articles cited, I didn't find inappropriate citations. The construction of the review is based on several very well described consequential paragraphs. Personally, I found very useful the synthesis of the aspects susceptible to new investigations (4. Future Research) and the paragraph on possible therapeutic consequences (5. Treatment Implications).  It’s certainly a review different from the typical "cliche" but in my opinion one of the best I could read.  I’m sure it’ll get numerous citations. I don’t have any relevant changes to suggest.

Author Response

Response to Reviewer 1

Journal: IJMS (ISSN 1422-0067)

Manuscript ID: ijms-2080046

Type: Review

Title: Amyotrophic Lateral Sclerosis Pathoetiology and Pathophysiology: Roles of Astrocytes, Gut microbiome and muscle interactions via the Mitochondrial melatonergic pathway, with disruption by Glyphosate-based herbicides

Author: George Anderson *

Section: Molecular Neurobiology

Reviewer 1

I read this review by George Anderson with great interest.  It deals the susceptibility to sporadic ALS determined by a dysregulation of the melatoninegic mitochondrial pathway that the Author hypothesizes may be caused by the ingestion of glyphosate-based herbicides with the diet. An important characterizing point of this hypothesis is to identify a fundamental role in the intestine and in particular in the altered intestinal microbiota. This is not a "neutral" review on the subject but is a real thesis that the author pursues with a careful and complete collection of all the literature in support of it. Despite the large number of articles cited, I didn't find inappropriate citations. The construction of the review is based on several very well described consequential paragraphs. Personally, I found very useful the synthesis of the aspects susceptible to new investigations (4. Future Research) and the paragraph on possible therapeutic consequences (5. Treatment Implications).  It’s certainly a review different from the typical "cliche" but in my opinion one of the best I could read.  I’m sure it’ll get numerous citations. I don’t have any relevant changes to suggest.

Response to Reviewer 1

Thank you for these encouraging comments.

Reviewer 2 Report

This is an exhaustive review about the integrative role of alterations in the mitochondrial melatonergic pathways and the already known or still unknown effects on some possible cells/tissues-targets in ALS.

The biochemical mechanisms are well detailed and the plausible already unknown mechanisms underpinning ALS pathobiology are fully discussed.

I would suggest some minor changes:

- Please put in italic font the genes’ names

- There are some typing errors, for instances in page 2 line 46: the square bracket in “[3” is not closed.

- Page 2, line 44, it would be more correct to write “approximately more than 30 genes”, instead of “other mutations”

- There are some abbreviations not specified at the end of the manuscript: pTDP43, AhR, mtSOD1, and others. Please specify all the abbreviations also in the text for a more readable manuscript.

- Page 3, line 115: the author stated that SSRI is a possible treatment for ALS, but to my current knowledge SSRI are not considered as a disease modifying therapy for ALS

- It would be useful to summarize the different mechanisms of GBH on GLIA, muscle and NMJ by using tables summarizing the corresponding references, in order to make easier the reading

- The paragraph 4 “Future Research” is very long and sometimes redundant. Is it possible to make it more discursive and fluid?

Author Response

Response to Reviewer 2

Journal: IJMS (ISSN 1422-0067)

Manuscript ID: ijms-2080046

Type: Review

Title: Amyotrophic Lateral Sclerosis Pathoetiology and Pathophysiology: Roles of Astrocytes, Gut microbiome and muscle interactions via the Mitochondrial melatonergic pathway, with disruption by Glyphosate-based herbicides

Author: George Anderson *

Section: Molecular Neurobiology

Reviewer 2

This is an exhaustive review about the integrative role of alterations in the mitochondrial melatonergic pathways and the already known or still unknown effects on some possible cells/tissues-targets in ALS.

The biochemical mechanisms are well detailed and the plausible already unknown mechanisms underpinning ALS pathobiology are fully discussed.

Response to Reviewer 2

Thank you for these encouraging comments.

I would suggest some minor changes:

- Please put in italic font the genes’ names

Response to Reviewer 2

Thank you for highlighting this. All gene names have now been put in italics.

- There are some typing errors, for instances in page 2 line 46: the square bracket in “[3” is not closed.

Response to Reviewer 2

Thank you for highlighting this. This has now been corrected.

- Page 2, line 44, it would be more correct to write “approximately more than 30 genes”, instead of “other mutations”

Response to Reviewer 2

Thank you for highlighting this. This has now been changed, as proposed.

- There are some abbreviations not specified at the end of the manuscript: pTDP43, AhR, mtSOD1, and others. Please specify all the abbreviations also in the text for a more readable manuscript.

Response to Reviewer 2

Acronyms used more than once have now been included in the list of abbreviations

- Page 3, line 115: the author stated that SSRI is a possible treatment for ALS, but to my current knowledge SSRI are not considered as a disease modifying therapy for ALS

Response to Reviewer 2

Although not formally recognized as disease-modifying, antidepressant (especially SSRI) use is common in ALS patients often in the hope of having some disease-modifying effects since Sandyk’s publication in 2006. 

Sandyk R. Serotonergic mechanisms in amyotrophic lateral sclerosis. Int J Neurosci. 2006 Jul;116(7):775-826. doi: 10.1080/00207450600754087. PMID: 16861147.

- It would be useful to summarize the different mechanisms of GBH on GLIA, muscle and NMJ by using tables summarizing the corresponding references, in order to make easier the reading

Response to Reviewer 2

The following table and legend have been added.

GBH effect

Glia

Gut

Muscles

NMJ

References

Shikimate pathway inhibition;

Decrease tryptophan, 5-HT, NAS, melatonin?

Altered gut bacteria patterning;

Altered 5-HT muscle regulation?

Altered MNs and NMJ 5-HT?

60,61,

increased gut permeability, circulating LPS

Reactivation, Inflammation, Increased YY1, lower EAATs

Altered gut bacteria interactions with mucosal immune system and enteric glia.

Muscle NF-kB driven inflammation

MNs apoptotic processes

62,64

GBH resistant gut bacteria

Lower butyrate/HDACi-linked epigenetic regulation. Altered fungal composition

Gut dysbiosis, HDACi

Lower butyrate/HDACi-linked epigenetic regulation. Mitochondrial dysfunction.

Lower butyrate/HDACi-linked epigenetic regulation.

62,63

Increased TNF-α and amyloid-β

Reactivation, Inflammation,

Gut permeability

Muscle damage. Lower muscle BDNF

MNs loss

73,75,99,111,123,124

Increase miR-34a, other miRNAs?

Lower TrkB-FL/TrkBT1 ratio

Gut permeability

Suboptimal muscle function

Lower TrkB-FL/TrkBT1 ratio

92,123,124

Suppresses  α7nAChR

Lost attenuation of glia and immune reactivity

Gut permeability

Heightened muscle inflammation

Heightened MNs and NMJ inflammation

93,95,98

Platelet dysregulation

BBB permeability

Gut permeability

Muscle atrophy

?

14,170-5

Table 1. Summarizes GBH effects in glia, gut, muscles and NMJ, with associated references. Abbreviations: 5-HT: 5-hydroxytryptamine (serotonin); α7nAChR: alpha 7 nicotinic acetylcholine receptor; BDNF: brain-derived neurotrophic factor; EAAT: excitatory amino acid transporter; GBH: glyphosate-based herbicides; HDACi: histone deacetylase inhibitor; LPS: lipopolysaccharide; MNs: motor neurons; NAS: N-acetylserotonin; NF-kB: nuclear factor kappa-light-chain-enhancer of activated B cells; NMJ: neuromuscular junction; TPH2: tryptophan hydroxylase; TrkB-FL: tyrosine receptor kinase B-full length; TrkB-T1: tyrosine receptor kinase B-truncated.

- The paragraph 4 “Future Research” is very long and sometimes redundant. Is it possible to make it more discursive and fluid?

Response to Reviewer 2

The ‘Future Research’ section has been trimmed to be more concise.
